# Enhanced Visible-Light Photocatalytic Activities of CeVO$_4$-V$_2$O$_3$ Composite: Effect of Ethylene Glycol

Yuxin Wang [1,*], Yinxiu Jin [1], Minghan Jia [2], Hu Ruan [3], Xuefen Tao [1], Xuefeng Liu [1], Guang Lu [4,*] and Xiaodong Zhang [5,*]

1 Institute of Applied Biotechnology, Taizhou Vocation & Technical College, Taizhou 318000, China
2 Zhejiang Jiuzhou Pharmaceutical Co., Ltd., Taizhou 318000, China
3 Taizhou Institute of Environmental Science Design and Research Co., Ltd., Taizhou 318000, China
4 School of Environmental and Safety Engineering, Liaoning Shihua University, Fushun 113001, China
5 School of Environment and Architecture, University of Shanghai for Science and Technology, Shanghai 200093, China
* Correspondence: wyx790914@aliyun.com (Y.W.); luguang20121101@126.com (G.L.); fatzhxd@126.com (X.Z.)

**Abstract:** CeVO$_4$-V$_2$O$_3$ composites were prepared by simple hydrothermal method, and the effects of ethylene glycol(EG) on the products were studied by XRD, N$_2$ adsorption–desorption, SEM, EDS, XPS, PL and UV-vis spectra. The characterization reveals a slight decrease in surface area and a slight enhancement of visible light absorption in the final sample, while the crystalline phase, morphology and separation efficiency of the collective carriers are severely affected by the EG. At the same time, the photocatalytic effect of CeVO$_4$-V$_2$O$_3$ composites was evaluated by the degradation rate of methylene blue (MB) under simulated visible light. The sample for 10 mL EG obtained the highest efficiency of 96.9%, while the one for 15 mL EG showed the lowest efficiency of 67.5% within 300 min. The trapping experiments and ESR experiment showed that the contribution of active species to the photocatalytic degradation of MB was $\cdot$OH > h$^+$ > $\cdot$O$^{2-}$ in descending order, and a possible degradation mechanism was proposed.

**Keywords:** ethylene glycol; CeVO$_4$-V$_2$O$_3$; methylene blue oxidation; visible light

## 1. Introduction

Photocatalytic technology is a green and economical solar-driven pollutant removal technology that can be carried out at normal temperatures and pressure. Using semiconductors and related materials as photocatalysts, we can carry out relatively compound chemical conversion under relatively simple conditions, which has unparalleled advantages in pollutant removal technology by directly using reducing agents or oxidizing agents. Photocatalytic technology provides an environmentally friendly and efficient transformation path for pollutant removal, leading many researchers to develop in a more cutting-edge research direction, and provides an efficient solution for carbon neutrality and sustainable development [1–6]. Nevertheless, TiO$_2$ cannot be excited when the wavelength is higher than 420 nm, which greatly limits its practical application in the visible light region. Therefore, research is devoted to developing a series of visible light-responsive photocatalysts to effectively utilize solar energy [7–11].

Cerium vanadate (CeVO$_4$), as a rare earth vanadate, has good physical and chemical properties and is widely used in the fields of batteries, semiconductors and catalysis. After vanadate is combined with rare earth metals such as Ce, La and Pr, its electrochemical performance, thermal stability, specific surface area and magnetism are obviously enhanced [12]. In view of the excellent physical and chemical properties of CeVO$_4$, many researchers use the precipitation method, microwave radiation method, sonochemical method and hydrothermal method to synthesize CeVO$_4$ [13–15]. In the absence of any catalyst and template, Xie et al. prepared tetragonal CeVO$_4$ with the same precursor by

hydrothermal method and ultrasonic methods, such as micro-rods, nanoparticles, nanorods and nanosheets [16]. Mosleh et al. reported a simple sonochemical method to synthesize $CeVO_4$ nanoparticles with the aid of ammonium metavanadate, using cerium (III) nitrate hexahydrate as the initial reagent and water as the solvent [17].

$CeVO_4$ has excellent catalytic performance, relatively stable chemical properties and a relatively simple preparation method, so it is a photocatalytic material with important research significance. Therefore, various micro/nanosized $CeVO_4$ samples have been prepared for use in the removal of organic compounds [18–21]. Phuruangrat et al. synthesized tetragonal zircon $CeVO_4$ photocatalytic particles by sol-gel method with tartaric acid as a complexing agent, which showed good activity in photocatalytic degradation of MB under ultraviolet light [22]. Liu et al. prepared egg yolk shell-like $CeVO_4$ microspheres composed of nanosheets by citric acid-assisted hydrothermal method, which realized the rare long cycle and high capacity of $CeVO_4$, and was considered as the next promising candidate negative electrode material for lithium-ion batteries [23]. However, considering the practical application, the photocatalytic activity of pure $CeVO_4$ still needs to be improved. The combination of $CeVO_4$ with other materials is considered to be an effective method to improve the separation efficiency of photo-generated carriers and improve photocatalytic performance [24]. Ma et al. prepared the flexible integrated composite of $CeVO_4$ and multi-walled carbon nanotubes (MWCNTs) by a simple hydrothermal technique. $CeVO_4$/MWCNTs composite can be applied to the detection of sulfadiazine residues in water samples of actual aquaculture [25]. Song et al. obtained the one-dimensional structure of AgNW@CeVO_4 composite photocatalyst by depositing $CeVO_4$ on the surface of a silver nanowire, which expanded the light absorption range of $CeVO_4$. In addition, compared with pure $CeVO_4$, AgNW@CeVO_4 composite photocatalyst showed excellent photocatalytic performance for the degradation of pollutants such as rhodamine B, MB and 4-nitrophenol under sunlight irradiation [26]. However, the above-mentioned combination methods were performed by adding additional materials, which undoubtedly increases the cost and complicates the process. Therefore, we try to study a simple and economical one-step preparation method; that is, cerium vanadate and vanadium anhydride can be obtained at the same time by controlling the reaction conditions, and they can form composites, thus improving the photocatalytic performance of $CeVO_4$.

In this paper, $CeVO_4$-$V_2O_3$ composites were prepared by a simple ethylene glycol (EG) assisted hydrothermal method. Because the solubility of ammonium vanadate is 4.8 at room temperature, and it is slightly soluble in water, adding organic solvent is a favorable dissolution method. Compared with ethanol, glycerol and other solvents, EG was chosen as the solvent. Moreover, the effects of EG addition on the crystal structure, morphology, separation efficiency of the recombination electron holes and photocatalytic activity have been studied in detail by XRD, $N_2$ adsorption–desorption, SEM, EDS, XPS, PL and UV-vis spectra. Trapping experiments and ESR are employed to study the active species in the photodegradation process. Based on the above results, the possible photodegradation mechanism of MB over $CeVO_4$-$V_2O_3$ composites is proposed.

## 2. Results and Discussion

### 2.1. Characterization and Analysis

The influence of EG on the crystal phase of the as-prepared samples was analyzed by X-ray powder diffraction (XRD). Figure 1 shows the XRD patterns of the composite samples. The four main peaks of these samples located at $2\theta$ = 18.3, 24.2, 32.4 and 46.5 are assigned to the (101), (200), (112) and (312) planes of $CeVO_4$, respectively. All the diffraction peaks of the products obtained from different EG addition were assigned to the tetragonal phase of $CeVO_4$ (JCPDS No. 12-0757). The XRD patterns also showed two main diffraction peaks at $2\theta$ = 28.6 and 36.9, corresponding to the (102) and (110) planes of the $V_2O_3$ phase (JCPDS No.01-071-0344).

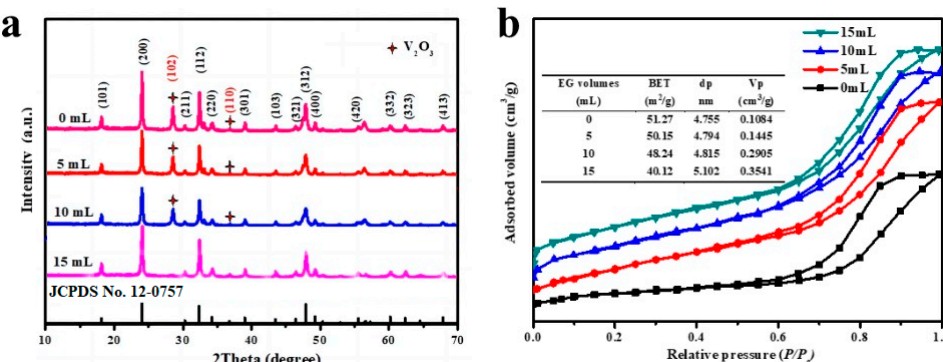

**Figure 1.** XRD pattern (**a**); N$_2$ adsorption–desorption isotherm curve of different samples (**b**).

It can also be seen from Figure 1a that the sample added with 15 mL EG only has the diffraction peak of tetragonal CeVO$_4$, and no impurity peak is detected, indicating that the CeVO$_4$-V$_2$O$_3$ composite has not been formed. In other samples, the diffraction peak corresponding to the V$_2$O$_3$ phase was detected, indicating the formation of the CeVO$_4$-V$_2$O$_3$ composites. Therefore, the addition of EG promotes the formation of the composites, but the excessive EG is not conducive to the formation of vanadic anhydride.

The specific surface area and porosity of the as-synthesized samples are determined by N$_2$ adsorption–desorption isotherms (Figure 1b). Obviously, the isotherms of all the samples were identified as type IV, with an H1 pattern hysteresis loop [27,28]. It could be deduced that as-prepared samples are members, and the specific surface area of obtained catalysts with additional amounts of 0, 5, 10 and 15 mL EG are 51.27, 50.15, 48.24 and 40.12 m$^2 \cdot$g$^{-1}$, respectively. These results show that the addition of EG leads to a slight decrease in specific surface area.

The morphology of as-prepared samples with different amounts of EG is investigated by SEM. As shown in Figure 2a, the product with no EG appears as a nanoparticle with an average diameter of about 230 nm. Adding 5 mL EG (as shown in Figure 2b), the product appears as a nano-tetrahedral bipyramid with an average diameter of about 550 nm. Upon increasing EG to 10 mL (as shown in Figure 2c), the product appears as microparticles with an average diameter of 1.0 μm, which is composed of several tetrahedrons and slices. Further increasing EG to 15 mL (as shown in Figure 2d), numerous leaf-like nano-slices are clustered together in groups to form a microsphere, caused by excessive EG probably. In addition, the average diameter increases to 1.6μm. The SEM results show that an increase in EG content leads to an increase in the average diameter of nanocomposites, which leads to a decrease in the specific surface area of nanocomposites. The chemical composition and element distribution of samples prepared in a solution containing 15 mL EG were studied by EDS. As shown in Figure 2e, the atomic ratio of O:V:Ce for the as-prepared sample is 66.46:17.82:15.72, which is larger than the theoretical stoichiometric ratio of 4:1:1 for O:V:Ce in CeVO$_4$.

X-ray photoelectron spectroscopy (XPS), as an important surface analysis technology, has the characteristics of simple preparation, no damage to the sample and can distinguish the chemical state information of elements, which has attracted more and more attention and use by researchers [29–31]. The chemical composition of the obtained catalysts is analyzed by XPS with 10 and 15 mL of EG addition. The survey scan spectra in Figure 3a is shown that both samples mainly contain Ce, V, C and O elements peaks (C 1s peak is assigned to the signal of the background hydrocarbon). As for the HR-XPS spectra of Ce 3d shown in Figure 3b, the Ce 3d$_{5/2}$ peaks locate at 881.1 and 884.9 eV and Ce 3d$_{3/2}$ peaks locate at 899.8 and 903.2 eV, indicating Ce in both samples mainly exists as Ce$^{3+}$ [32]. The HR-XPS spectra of V 2p resolve into two spin-orbit doublets (Figure 3c), characteristic of V$^{5+}$ and V$^{3+}$.The binding energies at 516.8 and 524.4 eV correspond to V 2p$_{3/2}$ and V 2p$_{1/2}$ of V$^{5+}$, and those at 515.4 and 522.6 eV can be attributed to V 2p$_{3/2}$ and V 2p$_{1/2}$ of V$^{3+}$. As for the HR-XPS spectra shown in Figure 3d, O 1s peaks are divided into the lattice oxygen

of CeVO$_4$ at 529.4 eV and O$^{2-}$ ions of V$_2$O$_3$ at 531.1 eV [33]. These results show that the final samples synthesized in solutions containing 10 and 15 mL of EG are CeVO$_4$-V$_2$O$_3$ composites, which are different from the XRD results. The possibility of such a result can be explained by the fact that V$_2$O$_3$ is distributed in amorphous form over the sample for 15 mL EG, which is not detectable by XRD.

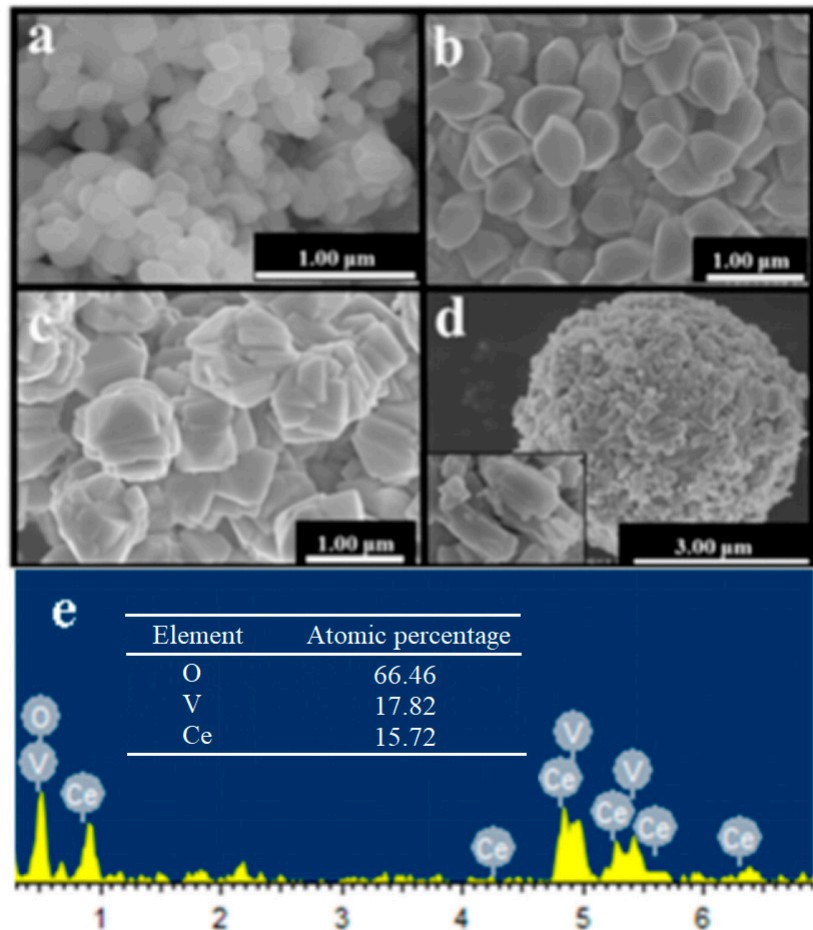

| Element | Atomic percentage |
|---------|-------------------|
| O | 66.46 |
| V | 17.82 |
| Ce | 15.72 |

**Figure 2.** SEM of CeVO$_4$-V$_2$O$_3$ prepared with EG addition amounts of 0 (**a**), 5 (**b**), 10 (**c**) and 15 mL (**d**) and EDS of CeVO$_4$-V$_2$O$_3$ prepared with EG addition amounts of 15 mL (**e**).

## 2.2. Photocatalytic Performance of CeVO$_4$-V$_2$O$_3$ Composites

Photoluminescence spectra (PL) can be used to determine the recombination of electrons and holes induced by light indirectly. Higher fluorescence intensity means higher electron and hole recombination efficiency [34,35]. Figure 4a shows the PL spectra of the CeVO$_4$-V$_2$O$_3$ composites. The emission bands centered at 512, 521, 556, 559 and 638 nm correspond to 5D0→7Fi (i = 0, 1, 2, 3, 4) electronic transitions of Ce$^{3+}$ ion, respectively [36]. This result indicates that the CeVO$_4$-V$_2$O$_3$ composites are excited by UV light (360 nm), transfer energy to Ce$^{3+}$, and then exhibit red emission through the f–f transition of Ce$^{3+}$ [37]. In addition, the strength of the CeVO$_4$-V$_2$O$_3$ sample without EG is high, which shows that the photo-generated electrons and holes generated by the sample under the condition of light excitation are easier to recombine and reduce the photocatalytic efficiency. The PL emission intensity of the sample synthesized in the EG precursor solution shows a dramatic weakness compared to that of the sample without EG. The intensity of the emission peak is the lowest for an added EG of 10 mL, indicating that the photo-generated electron and hole separation efficiency at the surface of this catalyst is maximally enhanced, and therefore, many photo-generated electrons and holes are generated.

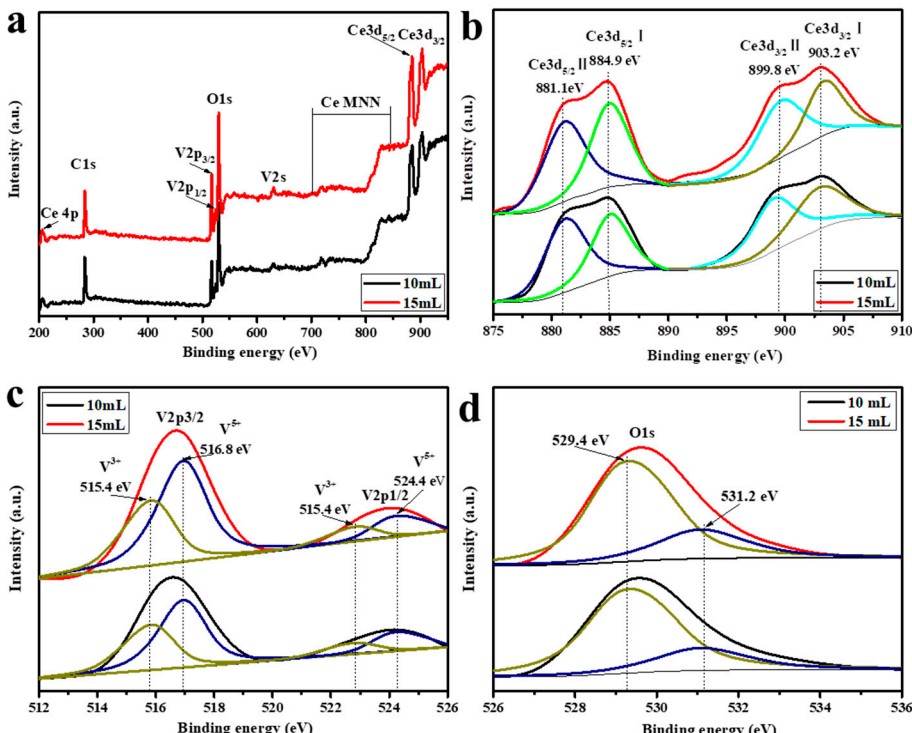

**Figure 3.** XPS results of the samples synthesized in the solutions containing 10 and 15 mL EG of survey (**a**), Ce 3d (**b**), V 2p (**c**) and O1s (**d**).

The optical properties of the as-synthesized samples were studied by UV-Vis absorption spectra (Figure 4b). The band gap of a semiconductor photocatalyst has a vital influence on its catalytic performance and determines the light absorption range in the process of photocatalysis. The band gap of the $CeVO_4$-$V_2O_3$ sample is calculated according to the following formula (Kubelka–Munk formula) [38,39]:

$$\alpha h\nu = A(h\nu - E_g)^2 \qquad (1)$$

where $\alpha$ stands for absorption coefficient, $\nu$ corresponds to the optical frequency, h is Planck constant and $E_g$ stands for the band gap of the sample. Figure 4b shows the curve of $(\alpha h\nu)^2$ versus $h\nu$, which is derived from the corresponding absorption spectra. Without EG, the four absorption peaks are located at 257, 353, 452 and 590 nm, respectively, and the energy gap is estimated to be 1.20 eV. When 5, 10 and 15 mL EG were added to the sample, the three absorption peaks were concentrated at 257, 452 and 568 nm, respectively, and the energy gap can be estimated as 1.03 eV. The results show that the addition of EG may enhance the absorption of visible light from the final products, indicating that they have an excellent UV-Vis response to visible light illumination. Comparatively speaking, the smaller the band gap of the composite, the less energy is needed for the electron transition reaction and the easier the photocatalytic reaction.

The catalytic activity of as-prepared $CeVO_4$-$V_2O_3$ photocatalysts was evaluated, and the effect of EG on the photocatalytic performance of the $CeVO_4$-$V_2O_3$ sample was also studied by simulating the photocatalytic conversion efficiency of MB under visible light ($\lambda > 420$ nm). Figure 4c shows the degradation efficiency of MB by photocatalyst in different irradiation times. Before photodegradation, the photocatalyst was dispersed in MB solution, and the adsorption–desorption equilibrium was completed in the dark for 60 min. It can be seen that the photocatalytic conversion efficiency of MB increases gradually with the increase of the amount of EG from 0 mL to 10 mL, but the photocatalytic conversion efficiency is the lowest when the amount of EG increases to 15 mL. It shows that the addition of EG is beneficial to the enhancement of the photocatalytic effect, but

it is not suitable to add too much. As can be observed, the photodegradation rate of MB can reach 96.9% within 300 min under the irradiation of visible light. However, the photocatalytic effect of the as-prepared $CeVO_4$-$V_2O_3$ composites is not as good as that of Ag nanowire@$CeVO_4$ heterostructure photocatalyst. Maybe introducing Ag can strengthen $O_2$ adsorption on the $CeVO_4$ surface, which advances the photocatalytic activity of $CeVO_4$ [26].

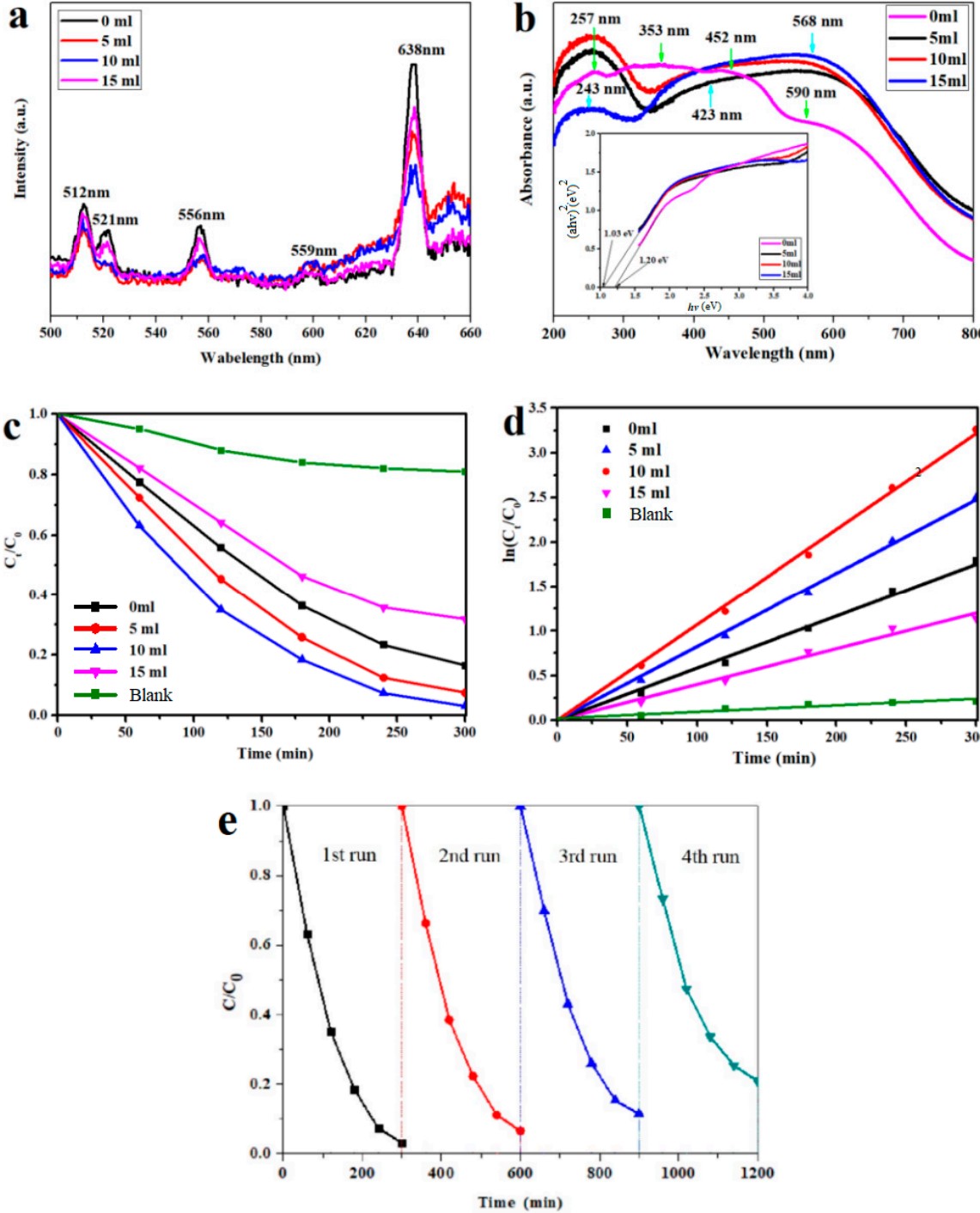

**Figure 4.** PL spectra (**a**), UV-vis absorption spectra (**b**), efficiency (**c**) and pseudo-first-order (**d**) of MB degradation using different $CeVO_4$-$V_2O_3$ samples under visible-light radiation. Cycling runs of MB photodegradation over $CeVO_4$-$V_2O_3$ samples with 10 mL EG (**e**).

In addition, based on the Langmuir–Hinshelwood (L-H) kinetic model, the photocatalysis degradation results conform to pseudo-first-order photocatalysis kinetics, and the corresponding reaction rate constants ($k$) of the different photocatalysts are calculated from the following equation [40,41]:

$$\ln (C_0/C) = kt \tag{2}$$

where $C_0$ corresponds to the initial concentration of MB solution, $C$ is the concentration of MB solution at $t$ minutes and $t$ stands for irradiation time. As can be clearly seen from Figure 4d, the $CeVO_4$-$V_2O_3$ photocatalyst prepared by adding 10 mL of EG has the highest $k$ value (0.0107 $min^{-1}$), which shows that adding 10 mL EG is helpful in optimizing the photocatalytic activity of $CeVO_4$-$V_2O_3$ for MB. The result is similar to the rate constants of 6 wt% $CeVO_4$/g-$C_3N_4$ [42].

Cycling runs of photodegradation of MB were performed to determine the stability of the $CeVO_4$-$V_2O_3$ sample with 10 mL EG. As shown in Figure 4e, the removal rates of MB by $CeVO_4$-$V_2O_3$ photocatalyst after four cycles decrease from 96.9% to 87.8%, indicating excellent photocatalytic stability. The result is consistent with the cycle stability of T-$CeVO_4$ [34].

As can be seen from the characterization results, EG slightly reduces the surface area and slightly improves the absorption of visible light in the final sample while severely affecting the crystalline phase, morphology and electron–hole separation efficiency. When the additive content of EG is increased from 0 to 10 mL, the final $CeVO_4$-$V_2O_3$ composite is formed, which improves the electron–hole separation efficiency and thus enhances the photocatalytic activity. Further increasing the amount of EG will form amorphous $V_2O_3$, and the nanosheets will gather together, which will reduce the electron–hole separation efficiency, thus reducing the photocatalytic activity.

## 2.3. Photocatalytic Mechanism of $CeVO_4$-$V_2O_3$ Composites

In order to explore the reaction mechanism of $CeVO_4$-$V_2O_3$ in the photocatalytic process, the trapping test of active substances was carried out, as shown in Figure 5. TEOA, BQ and IPA are trapping agents for the hole ($h^+$), superoxide radical ($\cdot O_2^-$) and hydroxyl radical ($\cdot OH$), respectively. During the photocatalytic oxidation $h^+$, $\cdot OH$ and $\cdot O_2^-$ usually separately or together act as the main radicals for destroying and mineralizing pollutants [43]. Thence, we added TEOA, BQ and IPA scavengers during the removal reaction of MB to remove $h^+$, $\cdot O_2^-$ or $\cdot OH$ species generated in the $CeVO_4$-$V_2O_3$ system, respectively [44]. Figure 5 shows the degradation rate of MB is reduced from 96.9% to 70.4% by the addition of TEOA, illustrating that $h^+$ plays a significant role in photocatalytic oxidation. Then, after BQ was added, the degradation efficiency of MB decreased from 96.9% to 86.8%, demonstrating that fewer $\cdot O_2^-$ participated in the photocatalytic reaction, which indicated that $\cdot O_2^-$ did not play a major role in the photocatalytic process. However, with the addition of IPA, the degradation rate of MB decreased significantly from 96.9% to 35.4%, indicating that additional $\cdot OH$ is involved in the photocatalysis process, which illustrated that $\cdot OH$ was an important active substance produced in the degradation process of MB. Thus, the order of influence of active species on the photocatalytic degradation of MB by $CeVO_4$-$V_2O_3$ is $\cdot OH > h^+ > \cdot O_2^-$. MB is mainly degraded by $\cdot O_2^-$ and $\cdot OH$ active species in $CeVO_4$-$V_2O_3$ samples under the irradiation of visible light.

Electron paramagnetic resonance (EPR) has been extensively used for the identification of free radicals that are generated from advanced oxidation processes (AOPs) so as to establish the reaction mechanism. According to the energy band theory of semiconductors, when the semiconductor is irradiated by light with energy equal to or greater than the forbidden band, electrons ($e^-$) in the valence band are excited to transition to the conduction band, and corresponding holes ($h^+$) are generated in the valence band. Photo-induced holes have strong electron acquisition ability, which can capture the electrons of water or hydroxyl adsorbed on the surface of photocatalysis particles and produce hydroxyl radicals. In ESR, 5,5-dimethyl-1-pyrroline N-oxide (DMPO) is usually used as a trapping

agent of ·OH, which can interact with photo-generated holes (h$^+$) or ·OH on the surface of photocatalysis to form a stable DMPO-OH adduct [45].

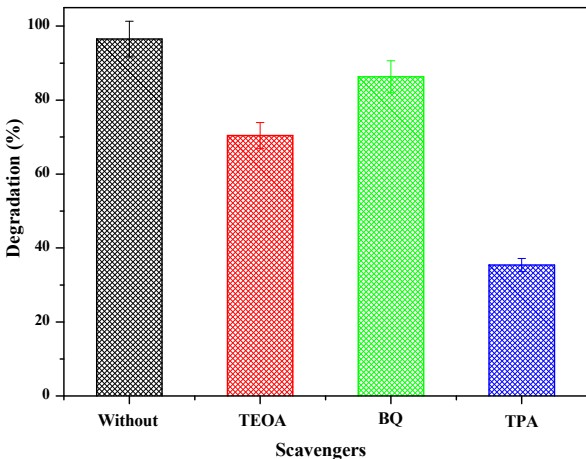

**Figure 5.** Trapping test of CeVO$_4$-V$_2$O$_3$ sample prepared with 10 mL EG.

To further identify the active species generated in the CeVO$_4$-V$_2$O$_3$ for 10 mL EG, the DMPO spin-trapping ESR technique was utilized to detect the production of ·OH and ·O$_2^-$. Figure 6a,b show the detection results of ·OH and ·O$_2^-$ generation, respectively. When it was not excited by visible light, ESR signals of DMPO-O$_2^-$ adducts and DMPO-OH adducts were not detected, indicating that no free radicals were produced in the CeVO$_4$-V$_2$O$_3$ sample. After being irradiated by visible light for 1 min, the characteristic peak of the DMPO-OH adduct can be clearly detected. Similarly, the typical signal of DMPO-O$_2^-$ adduct can also be detected, but the intensity is weaker than that of DMPO-OH, which is also consistent with the results of the active trapping experiment. The ESR signal intensity of spin adducts of DMPO-O$_2^-$ and DMPO-OH after 5 min of visible light irradiation is higher than that after 1 min of irradiation. Meanwhile, the signal intensity of DMPO-OH and DMPO-O$_2^-$ increase with light irradiation, which further confirms that ·OH and ·O$_2^-$ play a major role in promoting the degradation of MB. However, the intensities of DMPO-OH are obviously higher than that of DMPO-O$_2^-$, which indicates the CeVO$_4$-V$_2$O$_3$ for 10 mL EG produces additional ·OH active species. This result is consistent with the trapping test, which shows that ·OH was an important active substance produced in the degradation of MB.

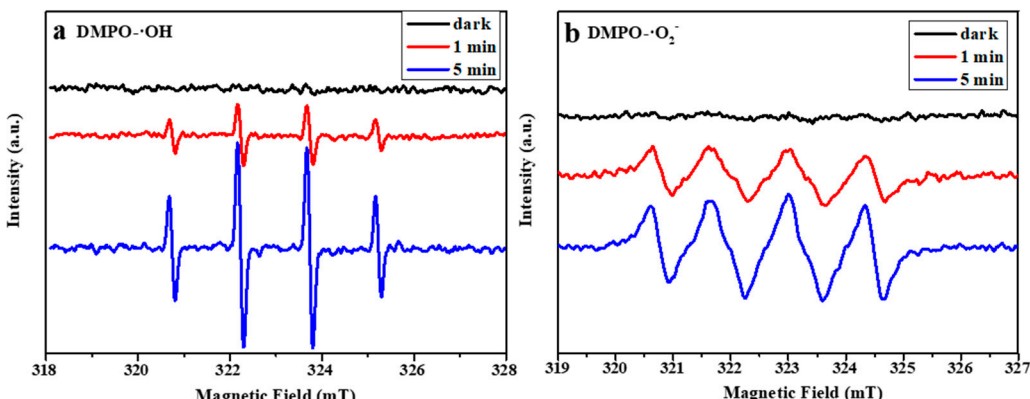

**Figure 6.** ESR spectra of the DMPO-·OH (**a**) and DMPO-·O$_2^-$ (**b**) for CeVO$_4$-V$_2$O$_3$ sample prepared with 10 mL EG.

Based on the above analysis, the tentative mechanism for the photocatalytic reaction of MB in CeVO$_4$-V$_2$O$_3$ samples prepared with 10 mL EG is proposed, as shown in Figure 7.

In the visible light irradiation, the photo-generated electrons in VB located in $V_2O_3$ and $CeVO_4$ migrate to CB in $V_2O_3$ and $CeVO_4$, respectively. After the coupling, the interface between $V_2O_3$ and $CeVO_4$ pushes the photo-generated electrons from the CB of $CeVO_4$ to the CB of $V_2O_3$. At the same time, the photo-generated holes are also transferred from the VB of $V_2O_3$ to the VB of $CeVO_4$. Therefore, faster electron transfer between $CeVO_4$ and $V_2O_3$ may lead to higher quantum efficiency and provide more photo-generated electrons for photocatalytic reactions. Then, the photo-generated electrons are captured by $O_2$ in water, resulting in superoxide radical $\cdot O^{2-}$, while the photo-generated holes are captured by the $OH^-$ or $H_2O$, resulting in $\cdot OH$. The oxidizability of $\cdot OH$, $\cdot O^{2-}$ and $h^+$ is enough to effectively degrade MB into $CO_2$ and $H_2O$.

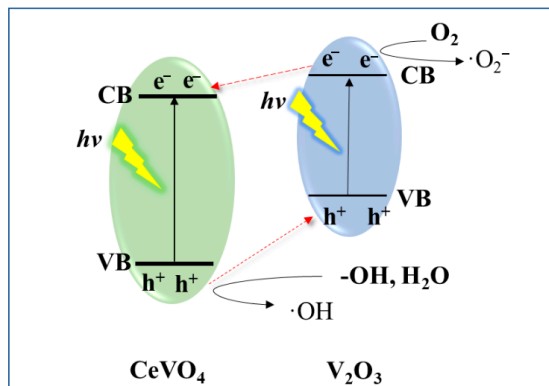

**Figure 7.** The photocatalytic mechanism of $CeVO_4$-$V_2O_3$ sample prepared with 10 mL EG.

## 3. Materials and Methods

### 3.1. Materials

All reagents used in the study, including ammonium metavanadate, ethylene glycol, cerium nitrate hexahydrate, ethylene glycol and anhydrous ethanol, are of analytical grade. They are purchased from Aladdin's company and used as received without further purification. All the solutions were prepared with deionized water obtained from a PURE ROUP 30 water purification system.

### 3.2. Synthesis of CeVO$_4$-V$_2$O$_3$ Composite

Typically, 2.17 g $Ce(NO_3)_3 \cdot 6H_2O$ and 0.59 g $NH_4VO_3$ were dispersed in 60 mL of deionized water at 90 °C under vigorous stirring for 1 h. Subsequently, 0–15 mL EG was added to the above suspension under stirring for an additional 1 h. Afterward, the obtained mixture was transferred into a Teflon-lined steel autoclave of 100 mL capacity, which was kept in an oven at 160 °C for 5 h. After naturally cooling in air, the obtained precipitates were alternately washed with deionized water and absolute ethanol three times and then dried at 100 °C for 12 h and calcined at 300 °C for 5 h (in $N_2$ atmosphere). A schematic representation of the synthesis of the $CeVO_4$-$V_2O_3$ composite is shown in Figure 8.

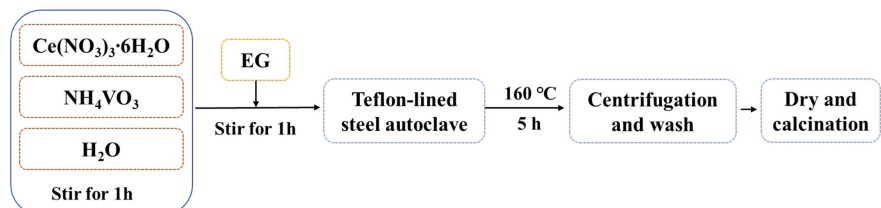

**Figure 8.** The schematic representation of synthesis of $CeVO_4$-$V_2O_3$ composite.

### 3.3. Characterization of CeVO$_4$-V$_2$O$_3$ Composite

The crystalline phase of the products was characterized by X-ray Diffractometer (Bruker Advance D-8, Billerica, MA, USA) with PIXel3D detector using CuKa radiation.

The diffraction patterns were matched with that of the recorded standards in JCPDS. The materials' surface morphology was studied with the aid of scanning electron microscopy-SEM (Hitachi S-4300 Field Emission SEM, Tokyo, Japan). The absorption spectra were determined by UV-Vis DRS (UV-2550), and the emission spectra were measured by photoluminescence spectra (PL, Spex Fluorolog-3, Metuchen, NJ, USA). The surface area was studied by $N_2$ adsorption–desorption (ASAP 2020). The elemental components and bonding energies of the synthesized samples were measured on X-ray photoelectron spectroscopy (XPS, VG Multilab 2000, London, UK).

### 3.4. Photocatalytic Performance Experiment

By studying the degradation of MB aqueous solution under visible light irradiation, the photocatalytic activities of $CeVO_4$-$V_2O_3$ composites with the addition of 0, 5, 10 and 15 were evaluated. A 500 W Xe discharge lamp with a 420 nm cut-off filter, equipped with a circulating water source and a 50 mg/L concentration of the dye, was prepared and used for the study. For each measurement, 50 mg of the catalyst was suspended in MB aqueous solution and stirred vigorously in the dark for 60 min to ensure the establishment of adsorption equilibrium between the catalyst surface and the dye. Then, the suspension was exposed to a 500 W Xe discharge lamp with a 420 nm cut-off filter and equipped with a circulating water source and was continuously stirred for 300 min. During this process, 5 mL of the suspension in the reaction system was taken out for testing at 60, 120, 180, 240 and 300 min, respectively. The resulting supernatants were examined using a UV-1100 spectrophotometer, and the absorbance of the MB solution was measured at its characteristic absorption wavelength of 664 nm. The degradation curve of the dye was studied by absorption spectrum. Use the formula in Equation (3) to calculate the degraded MB concentration.

$$\text{Degradation } (\%) = \frac{C_0 - C_i}{C_0} \times 100\% = \frac{A_0 - A_i}{A_0} \times 100 \ (\%) \tag{3}$$

where $C_i$ is the concentration of MB at specific time; $C_0$ denotes the initial concentration of MB; $A_i$ is the absorbance of MB at different times and $A_0$ denotes the blank absorbance of the original MB solution.

### 3.5. Photocatalytic Mechanism Tests

The trapping tests are similar to the photodegradation experiment. The difference is that 1 mmol of isopropanol (IPA), triethanolamine (TEOA) or benzoquinone (BQ) is added into the reaction solution before irradiation, and the degradation rate of MB after irradiation for 300 min is analyzed. Species ($h^+$, $\cdot O_2^-$ and $\cdot OH$) formed in the photodegradation process were studied with electron spin resonance (ESR, Bruker E500, Billerica, MA, USA) by adding 5,5-dimethyl-1-pyrroline N-oxide (DMPO, >99.0%) into ultrapure water and methanol, respectively.

## 4. Conclusions

The effects of EG on the crystalline phase, surface area, morphology, chemical composition, electron–hole separation efficiency and optical properties of the final product have been studied using XRD, $N_2$ adsorption–desorption, SEM, XPS, PL and DRS. The results show that a proper amount of EG can form $CeVO_4$-$V_2O_3$ composites. With the addition of EG, the specific surface area of the composites decreased slightly, the average particle size increased, and the visible light absorption increased. At the same time, the photocatalytic removal effect of $CeVO_4$-$V_2O_3$ composites on MB shows that adding moderate EG into the precursor solution (0–10 mL) can improve the electron–hole separation efficiency of the final products, thus increasing the photocatalytic activity (83.6% to 96.9%). However, excessive EG (15 mL) makes the nanosheets gather together to form aggregates, which reduces the separation efficiency of electron holes, thus reducing the photocatalytic activity (67.5%). According to the trapping experiments and ESR experiments, $h^+$ and $\cdot OH$ played

a more significant part in the photocatalytic degradation of MB than $\cdot O_2^-$, and the order of influence of active species on the photodegradation of MB over $CeVO_4$-$V_2O_3$ sample prepared with 10 mL EG is $\cdot OH > h^+ > \cdot O_2^-$. Finally, the mechanism of photocatalytic degradation of MB by the $CeVO_4$-$V_2O_3$ composites was discussed. The coupling of $V_2O_3$ and $CeVO_4$ realizes the effective separation and faster transfer of photo-generated electrons and holes and leads to higher quantum efficiency. When they are captured by $O_2$ and $OH^-$ in water, active species, $\cdot O_2^-$ and $\cdot OH$, are generated. The oxidizability of $\cdot OH$, $\cdot O_2^-$ and $h^+$ is enough to effectively degrade MB into $CO_2$ and $H_2O$.

**Author Contributions:** Data curation, Y.W.; formal analysis, Y.W., M.J. and X.T.; investigation, M.J. and H.R.; methodology, X.L.; resources, Y.W., G.L. and X.Z.; writing—original draft, Y.W.; writing—review and editing, Y.W., Y.J. and X.T. All authors have read and agreed to the published version of the manuscript.

**Funding:** This work was supported by the "School Enterprise Cooperation Project" for Domestic Visiting Engineers of Colleges and Universities (No. FG2022260), the project of Taizhou Science and technology planning (2101gy32) and the cultivating project of Taizhou vocational and technical college (2021PY04).

**Informed Consent Statement:** Written informed consent has been obtained from the patient(s) to publish this paper.

**Data Availability Statement:** Not applicable.

**Conflicts of Interest:** The authors declare that they have no known competing financial interest or personal relationship that could have appeared to influence the work reported in this paper.

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
