# Peer review of "Enhanced Visible-Light Photocatalytic Activities of CeVO4-V2O3 Composite: Effect of Ethylene Glycol"

_catalysts, doi:10.3390/catal13040659_

Round 1

Reviewer 1 Report

This paper describes the Enhanced visible-light photocatalytic activities of CeVO4 -V2O3 composite: Effect of ethylene glycol. After reviewing it, I think it can be consider to publish if the following issues are solved:

1. For the introduction part, the recent achievements of this kind of catalysts and the necessary of this work should be clarified.

2. For the xrd patterns, the pdf card of V2O3 and CeVO4 should be added.

3. For the figure 2, the element mapping should be added.

4. How about the stability of this kind of catalysts? The stability test should be added.

5.  Some relevant papers can be cited:DOI 10.1002/batt.202200434ï¼›ACS NANO 13 (3) , pp.3600-3607ï¼›doi.org/10.1021/acsnano.7b01152ï¼›Separation and Purification Technology 303 (2022) 122288; https://doi.org/10.1016/j.jcis.2022.09.014.

Author Response

This paper describes the Enhanced visible-light photocatalytic activities of CeVO4 -V2O3 composite: Effect of ethylene glycol. After reviewing it, I think it can be consider to publish if the following issues are solved:

  1. For the introduction part, the recent achievements of this kind of catalysts and the necessary of this work should be clarified.

Answers:

We appreciate your comments and are pleased to receive them, and we have revised the introduction within the article. The details are as follows:

(Page 2, line 56-58: CeVO4 has excellent catalytic performance, relatively stable chemical properties and relatively simple preparation method, so it is a photocatalytic material with important research significance.

 Page 2, line 78-83: However, the above-mentioned combination methods were done by adding additional materials, which undoubtedly increases the cost and complicates the process. Therefore, we try to study a simple and economical one-step preparation method, that is, cerium vanadate and vanadium anhydride can be obtained at the same time by controlling the reaction conditions, and they can form composites, thus improving the photocatalytic performance of CeVO4.

Page 2, line 85-88: Because ammonium vanadate is insoluble in water, ethanol, ether and ammonium chloride at room temperature, it is necessary to choose a soluble organic solvent. According to the comparative experiment results of glycerol, nitric acid and other solvents, we chose ethylene glycol as the solvent. )

  1. For the xrd patterns, the pdf card of V2O3 and CeVO4 should be added.

Answers:

  We are very grateful to the reviewers for your suggestions. we have added relevant content in the Figure 1a. The details are as follows:

Figure 1. XRD pattern (a); N2 adsorption-desorption isotherm curve of different samples (b).

  1. For the figure 2, the element mapping should be added.

Answers:

We greatly appreciate reviewer’s suggestion. It is true that elemental mapping is a very intuitive means of elemental analysis, but the as-prepared composites contain lighter elements and the graphic resolution obtained by ordinary EDS mapping is not high, so it was not included. However, each element of the as-prepared composites is analyzed in detail in the XPS characterization part, which can be effectively explained.

  1. How about the stability of this kind of catalysts? The stability test should be added.

Answers:

  We are very grateful to the reviewers for your suggestions. we have reviewed and added relevant content to the text. The details are as follows:

(Page 6, line 225-229: Cycling runs of photodegradation of MB were performed to determine the stability of CeVO4-V2O3 sample with 10ml EG. As shown in Figure 4f, the removal rates of MB by  CeVO4-V2O3 photocatalyst after four cycles decrease from 96.9% to 87.8%, indicating  excellent photocatalytic stability. The result is consistent with the cycle stability of T-CeVO4 [34].

  1. Some relevant papers can be cited:DOI 10.1002/batt.202200434;ACS NANO 13 (3) , pp.3600-3607;org/10.1021/acsnano.7b01152;Separation and Purification Technology 303 (2022) 122288; https://doi.org/10.1016/j.jcis.2022.09.014.

Answers:

  We are very grateful to the reviewers for your suggestions. Based on the references suggested by the reviewers, we have reviewed and added relevant content to the text. The relevant references that were added are shown below:

([5] Guo, W.; Sun, W.; Lv, L.; Kong, S.; Wang, Y. Microwave-assisted morphology evolution of Fe-based metal–organic frameworks and their derived Fe2O3 nanostructures for li-Ion storage.  ACS Nano. 2017, 11, 4198−4205.

[6] Chen,X.; Zhang, Hang.; Ci, C.; Sun, W.; Wang, Yong. Few-Layered Boronic Ester Based Covalent Organic Frameworks/Carbon Nanotube Composites for High-Performance K‑Organic Batteries.: ACS Nano. 2019, 13, 36003607.

[29] Zhang, B.; Wu, M.; Zhang, L.; Xu, Y.; Hou, W.; Guo, H.; Wang L. Isolated transition metal nanoparticles anchored on n-doped carbon nanotubes as scalable bifunctional electrocatalysts for efficient Znair batteries. J. Colloid Interface Sci. 2023, 629, 640648.

[30] Wang, H.; Zou, W.; Liu, C.; Sun, Y.; Xu, Y.; Sun, W.; Wang, Yong. β-Ketoenamine-Linked Covalent Organic Framework with Co Intercalation: Improved Lithium-Storage Properties and Mechanism for High-Performance Lithium-Organic Batteries. Batteries & Supercaps. 2023, e202200434.

[38] Xie, Y.; Zhou, Y.; Gao, C.; Liu, L.; Zhang, Y.; Chen, Y.; Shao, Yi. Construction of AgBr/BiOBr S-scheme heterojunction using ion exchange strategy for high-efficiency reduction of CO2 to CO under visible light. Sep. Purif. Technol. 2022, 303, 122288.)

Reviewer 2 Report

The manuscript entitled Enhanced visible-light photocatalytic activities of CeVO4-V2O3 composite: Effect of ethylene glycol by Yuxin Wang et al. describes the preparation of four materials and their characterization. Additionally, the ability of the materials to remove methylene blue using visible light was tested.

The paper is generally well written in terms of style, it fits the scope of the journal Catalysts.

However, some major changes are required in the manuscript:

1.      The most significant issue: the manuscript is written as a technical report, listing the results and analyses performed, however it lacks the scientific approach. I understand that new materials were prepared but the only information I can find is “what” changed and not “why did it change”. The Results and Discussion part lacks some insight into that. Please improve the discussion throughout the article.

2.      What was the aim of the study? Why was EG chosen as the addition to the preparation procedure? Currently it feels random. Please provide some explanation and add to the introduction a fragment on EG in synthesis of materials.

3.      There is no description of the experiments with radical quenchers in the materials and methods section.

4.      Why were materials with 0 and 5 mL EG addition not tested by XPS?

5.      Photocatalytic experiments: were the kinetic constants corrected by subtracting the photobleaching of MB? Was MB photobleaching tested? Please add the error bars in figures 4c, 4d and 5. Also, please put the obtained photocatalytic efficiency in context by comparing the measured effects to other photocatalytic materials.

Less significant remarks:

6.      The description of figure 5 does not provide sufficient information to be self-explanatory.

7.      The sentences in lines 274-275 and 350-351 – the mineralization of the samples was not tested, so no such statement should be used.

8.      Were the materials tested for their recyclability?

Author Response

The manuscript entitled Enhanced visible-light photocatalytic activities of CeVO4-V2O3 composite: Effect of ethylene glycol by Yuxin Wang et al. describes the preparation of four materials and their characterization. Additionally, the ability of the materials to remove methylene blue using visible light was tested.

The paper is generally well written in terms of style, it fits the scope of the journal Catalysts.

However, some major changes are required in the manuscript:

  1. The most significant issue: the manuscript is written as a technical report, listing the results and analyses performed, however it lacks the scientific approach. I understand that new materials were prepared but the only information I can find is “what” changed and not “why did it change”. The Results and Discussion part lacks some insight into that. Please improve the discussion throughout the article.

We appreciate your comments and are pleased to receive them, and we have improved the discussion throughout the article.. The details are as follows:

(Page 3, line 111-123: Therefore, the addition of EG promotes the formation of the composites, but the excessive EG is not conducive to the formation of vanadic anhydride.

Page 4, line 142-145: X-ray photoelectron spectroscopy (XPS) , as an important surface analysis technology, has the characteristics of simple preparation, no damage to the sample, and can distinguish the chemical state information of elements, which has attracted more and more attention and use by researchers[29-30].

Page 4, line 156-159: These results show that the final samples synthesized in solutions containing 10 and 15 mL of EG are CeVO4-V2O3 composites, which is different from the XRD results. The possibility of such a result can be explained by the fact that V2O3 is distributed in amorphous form over the sample for 15 mL EG, which is not detectable by XRD.

Page 5, line 181-184: The optical properties of the as-synthesized samples were studied by Uv-vis absorption spectra (Figure 4b). The band gap of semiconductor photocatalyst has a vital influence on its catalytic performance, and determines the light absorption range in the process of photocatalysis.

Page 6, line 190-193: Without EG, the four absorption peaks are located at 257, 353, 452 and 590 nm, respectively, and the energy gap is estimated to be 1.20 eV. When 5 , 10 and 15 mL EG were added to the sample, the three absorption peaks were concentrated at 257, 452 and 568 nm, respectively, and the energy gap can be estimated as 1.03 eV.

Page 6, line 196-197: Comparatively speaking, the smaller the band gap of the composite, the less energy needed for the electron transition reaction, and the easier the photocatalytic reaction.

Page 6, line 210-213: However, the photocatalytic effect of the as-prepared CeVO4-V2O3 composites is not as good as that of Ag nanowire@CeVO4 heterostructure photocatalyst[26]. Maybe introducing Ag can strengthen O2 adsorption on CeVO4 surface, which advances the photocatalytic activity of CeVO4.

Page 6, line 225-229: Cycling runs of photodegradation of MB were performed to determine the stability of CeVO4-V2O3 sample with 10ml EG. As shown in Figure 4f, the removal rates of MB by  CeVO4-V2O3 photocatalyst after four cycles decrease from 96.9% to 87.8%, indicating  excellent photocatalytic stability. The result is consistent with the cycle stability of T-CeVO4 [34].

Page 7, line 244-247: In order to explore the reaction mechanism of CeVO4-V2O3 in photocatalytic process, the trapping test of active substances was carried out, as shown in Figure 5. TEOA, BQ and IPA are trapping agents for hole (h+), superoxide radical (∙O2) and hydroxyl radical (∙OH) respectively.

Page 8, line 259-262: Thus, the order of influence of active species on the photocatalytic degradation of MB by CeVO4-V2O3 is ∙OH> h+ > ∙O2. MB is mainly degraded by ∙O2 and ∙OH active species in CeVO4-V2O3 samples under the irradiation of visible light.

Page 9, line 292-293: This result is consistent with the trapping test, which shows that ∙OH was an important active substance produced in the degradation of MB.)

  1. What was the aim of the study? Why was EG chosen as the addition to the preparation procedure? Currently it feels random. Please provide some explanation and add to the introduction a fragment on EG in synthesis of materials.

Answers:

We appreciate your comments and are pleased to receive them, and we have revised the introduction within the article. The details are as follows:

(Page 2, line 56-58: CeVO4 has excellent catalytic performance, relatively stable chemical properties and relatively simple preparation method, so it is a photocatalytic material with important research significance.

 Page 2, line 78-83: However, the above-mentioned combination methods were done by adding additional materials, which undoubtedly increases the cost and complicates the process. Therefore, we try to study a simple and economical one-step preparation method, that is, cerium vanadate and vanadium anhydride can be obtained at the same time by controlling the reaction conditions, and they can form composites, thus improving the photocatalytic performance of CeVO4.

Page 2, line 85-89: Because ammonium vanadate is insoluble in water, ethanol, ether and ammonium chloride at room temperature, it is necessary to choose a soluble organic solvent. According to the comparative experimental results of solvent selection in the previous period, we chose ethylene glycol as the solvent. )

  1. There is no description of the experiments with radical quenchers in the materials and methods section.

Answers:

We are very grateful to the reviewers for this suggestion. We have added the experiments in the materials and methods section. The details are as follows:

(Page 11, line 362-369:

3.5. Photocatalytic mechanism tests  

The trapping tests is similar to the photodegradation experiment. The difference is that 1 mmol of isopropanol (IPA), triethanolamine (TEOA) or benzoquinone (BQ) are added into the reaction solution before irradiation, and the degradation rate of MB after irradiation for 300 min is analyzed. Species (h+, ∙O2 and ∙OH) formed in the photodegradation process were studied with electron spin resonance (ESR, Bruker E500) by adding 5,5-dimethyl-1-pyrroline N-oxide (DMPO, >99.0%) into ultrapure water and methanol, respectively.)

  1. Why were materials with 0 and 5 mL EG addition not tested by XPS?

Answers:

We greatly appreciate reviewer’s suggestion. In fact, it makes sense to characterize all samples. However, according to the XRD results, the CeVO4-V2O3 composites were obtained by adding 0, 5 and 10 ml EG, while the single CeVO4 was obtained by adding 15 ml EG. Therefore, we selected the composite with good photocatalytic degradation effect by adding 10 ml EG and the single sample by adding 15 ml EG for XPS characterization.

  1. Photocatalytic experiments: were the kinetic constants corrected by subtracting the photobleaching of MB? Was MB photobleaching tested? Please add the error bars in figures 4c, 4d and 5. Also, please put the obtained photocatalytic efficiency in context by comparing the measured effects to other photocatalytic materials.

Answers:

We greatly appreciate reviewer's suggestion. The photobleaching of MB was tested, and related data information was added, as shown in Figures 4c and 4d. According to the reviewer's suggestion, we added the error bars in Figure 5. However, because there are many points and lines in Figures 4c and 4 d, if error bars are added, the figures will be confused, so there is no error bars added in these two figures. Also, we compared the photocatalytic effect of the as-prepared CeVO4-V2O3 composites with that of other photocatalytic materials. The details are as follows:

(Page 6, line 210-213: However, the photocatalytic effect of the as-prepared CeVO4-V2O3 composites is not as good as that of Ag nanowire@CeVO4 heterostructure photocatalyst[26]. Maybe introducing Ag can strengthen O2 adsorption on CeVO4 surface, which advances the photocatalytic activity of CeVO4.

Figure 4. PL spectra(a), UV-vis absorption spectra(b), efficiency(c) and pseudo-first-order(d) of MB degradation using different CeVO4-V2O3 samples under visible-light radiation. Cycling runs of MB photodegradation over CeVO4-V2O3 samples with 10ml EG(f).

Figure 5. Trapping test of CeVO4-V2O3 sample prepared with 10mL EG.

Less significant remarks:

  1. The description of figure 5 does not provide sufficient information to be self-explanatory.

Answers:

We greatly appreciate reviewer’s suggestion. We have added the description of figure 5. The details are as follows:

(Page 7, line 244-247: In order to explore the reaction mechanism of CeVO4-V2O3 in photocatalytic process, the trapping test of active substances was carried out, as shown in Figure 5. TEOA, BQ and IPA are trapping agents for hole (h+), superoxide radical (∙O2) and hydroxyl radical (∙OH) respectively.)

  1. The sentences in lines 274-275 and 350-351 – the mineralization of the samples was not tested, so no such statement should be used.

Answers:

We greatly appreciate reviewer’s suggestion. We have replaced the inappropriate expressions as follows:

(Page 9, line 297-298: Based on the above analysis, the tentative mechanism for photocatalytic reaction of MB in CeVO4-V2O3 samples prepared with 10mL EG is proposed, as shown in Figure 7.

Page 12, line 377-378: Meanwhile, the photocatalytic effect of obtained CeVO4-V2O3 composites were identified by the removal ratio of MB.)

  1. Were the materials tested for their recyclability?

Answers:

  We are very grateful to the reviewers for your suggestions. we have reviewed and added relevant content to the text. The details are as follows:

(Page 6, line 225-229: Cycling runs of photodegradation of MB were performed to determine the stability of CeVO4-V2O3 sample with 10ml EG. As shown in Figure 4f, the removal rates of MB by  CeVO4-V2O3 photocatalyst after four cycles decrease from 96.9% to 87.8%, indicating  excellent photocatalytic stability. The result is consistent with the cycle stability of T-CeVO4 [34].

Figure 4. PL spectra(a), UV-vis absorption spectra(b), efficiency(c) and pseudo-first-order(d) of MB degradation using different CeVO4-V2O3 samples under visible-light radiation. Cycling runs of MB photodegradation over CeVO4-V2O3 samples with 10ml EG(f).

Round 2

Reviewer 1 Report

It is well revised and can be accepted now.

Author Response

thanks

Reviewer 2 Report

Thank you for the changes made in the new version of the manuscript. The manuscript significantly improved and can be accepted for publishing.

There are however some errors in the introduced text that seem illogical, i.e. solubility in ammonium cloride at room temperature (lines 83-84) or naming nitric acid an organic solvent (line 85). Please read the article carefully and correct all the technical mistakes that will not be corrected by the MDPI English editing team.

Author Response

Thank you for the changes made in the new version of the manuscript. The manuscript significantly improved and can be accepted for publishing.

There are however some errors in the introduced text that seem illogical, i.e. solubility in ammonium cloride at room temperature (lines 83-84) or naming nitric acid an organic solvent (line 85). Please read the article carefully and correct all the technical mistakes that will not be corrected by the MDPI English editing team.

Answers:

We appreciate your comments and are pleased to receive them, and we have revised the introduction within the article. The details are as follows:

(Page 1, line 30-38: which can be carried out at normal temperature and pressure. Using semiconductors and related materials as photocatalysts, relatively compound chemical conversion can be carried out under relatively simple conditions, which has unparalleled advantages in pollutant removal technology by directly using reducing agents or oxidizing agents. Photocatalytic technology provides an environmentally- friendly and efficient transformation path for pollutant removal, leading many researchers to develop in a more cutting-edge research direction, and provides an efficient solution for carbon neutrality and sustainable development.[1-6]. Nevertheless, TiO2 can’t be excited when the wavelength is higher than 420 nm,

Page 2, line 83-87: In this paper, CeVO4-V2O3 composites were prepared by a simple ethylene glycol(EG) assisted hydrothermal method. Because the solubility of ammonium vanadate is 4.8 at room temperature, and it is slightly soluble in water, adding organic solvent is a favorable dissolution method. Compared with ethanol, glycerol and other solvents, EG was chosen as the solvent.

Page 2, line 90-92: Trapping experiments and ESR are employed to study the active species in photodegradation process. Based on the above results, the possible photodegradation mechanism of MB over CeVO4-V2O3 composites is proposed.

Page 6-7, line 231-233: Further increasing the amount of EG will form amorphous V2O3, and the nanosheets will gather together, which will reduce the electron-hole separation efficiency, thus reducing the photocatalytic activity.

Page 11, line 370-378: The results show that a proper amount of EG can form CeVO4-V2O3 composites. With the addition of EG, the specific surface area of the composites decreased slightly, the average particle size increased, and the visible light absorption increased. At the same time, the photocatalytic removal effect of CeVO4-V2O3 composites on MB shows that adding moderate EG into the precursor solution (0-10 mL) can improve the electron-hole separation efficiency of the final products, thus increasing the photocatalytic activity (83.6% to 96.9%). However, excessive EG (15 mL) makes the nanosheets gather together to form aggregates, which reduces the separation efficiency of electron holes, thus reducing the photocatalytic activity (67.5%). )